# In Vitro Hypocholesterolemic Effect of Coffee Compounds

**DOI:** 10.3390/nu12020437

**Published:** 2020-02-09

**Authors:** Filipe Manuel Coreta-Gomes, Guido R. Lopes, Cláudia P. Passos, Inês M. Vaz, Fernanda Machado, Carlos F. G. C. Geraldes, Maria João Moreno, Laura Nyström, Manuel A. Coimbra

**Affiliations:** 1LAQV-REQUIMTE, Chemistry Department, University of Aveiro, 3810-193 Aveiro, Portugal; guido@ua.pt (G.R.L.); cpassos@ua.pt (C.P.P.); inesmsilva@ua.pt (I.M.V.); fernandamachado@ua.pt (F.M.); mac@ua.pt (M.A.C.); 2Coimbra Chemistry Center, University of Coimbra, Rua Larga Largo D. Dinis, 3004-535 Coimbra, Portugal; geraldes@uc.pt (C.F.G.C.G.); mmoreno@ci.uc.pt (M.J.M.); 3Department of Life Sciences, Faculty of Science and Technology, University of Coimbra, Calçada Martim de Freitas, 3000-456 Coimbra, Portugal; 4Chemistry Department, University of Coimbra, Faculty of Science and Technology, Rua Larga Largo D. Dinis, 3004-535 Coimbra, Portugal; 5ETH Zurich, Institute of Food, Nutrition and Health, Schmelzbergstrasse 9, CH-8092 Zurich, Switzerland; laura.nystroem@hest.ethz.ch

**Keywords:** coffee, polysaccharides, hypocholesterolemic effect, sequestration, bioaccessibility, NMR, lipid, bile salts, cholesterol

## Abstract

(1) Background: Cholesterol bioaccessibility is an indicator of cholesterol that is available for absorption and therefore can be a measure of hypocholesterolemic potential. In this work, the effect of commercial espresso coffee and coffee extracts on cholesterol solubility are studied in an in vitro model composed by glycodeoxycholic bile salt, as a measure of its bioaccessibility. (2) Methods: Polysaccharide extracts from coffees obtained with different extraction conditions were purified by selective precipitation with ethanol, and their sugars content were characterized by GC-FID. Hexane extraction allowed us to obtain the coffee lipids. Espresso coffee samples and extracts were tested regarding their concentration dependence on the solubility of labeled ^13^C-4 cholesterol by bile salt micelles, using quantitative ^13^C NMR. (3) Results and Discussion: Espresso coffee and coffee extracts were rich in polysaccharides, mainly arabinogalactans and galactomannans. These polysaccharides decrease cholesterol solubility and, simultaneously, the bile salts’ concentration. Coffee lipid extracts were also found to decrease cholesterol solubility, although not affecting bile salt concentration. (4) Conclusions: Coffee soluble fiber, composed by the arabinogalactans and galactomannans, showed to sequester bile salts from the solution, leading to a decrease in cholesterol bioaccessibility. Coffee lipids also decrease cholesterol bioaccessibility, although the mechanism of action identified is the co-solubilization in the bile salt micelles. The effect of both polysaccharides and lipids showed to be additive, representing the overall effect observed in a typical espresso coffee. The effect of polysaccharides and lipids on cholesterol bioaccessibility should be accounted on the formulation of hypocholesterolemic food ingredients.

## 1. Introduction

Cholesterol is a critical player in many biological processes, such as modulating membrane structure and function, and as precursor of bile salts and essential hormones and vitamins [1,2,3,4,5]. The bile salts, on their turn, have an important action on the solubilization of hydrophobic molecules in the intestine and ultimately on their absorption. However, excess cholesterol leads to the development of atherosclerosis. The accumulation of cholesterol in the arteries, primarily in low-density lipoproteins (LDL), evolves into the formation of arteriosclerotic plaques and increase of blood pressure [6,7]. This condition results in a high prevalence of strokes and heart attacks with elevated mortality, disability indexes and high healthcare costs [8].

There are three main strategies targeting the reduction of cholesterol levels in the bloodstream by affecting the following: (i) the endogenous production of cholesterol; (ii) the absorption of cholesterol coming both from the diet and endogenous sources that is discharged from the bile after major meals and (iii) both processes [9,10,11]. 

The second strategy involves the reduction of cholesterol absorption through the intestine. One approach is to use cationic resins, such as colestipol, that have the capacity to sequester bile salts, leading to a reduction of their concentration in the intestinal lumen, lowering the cholesterol solubility, therefore limiting its absorption through the intestine [12,13,14]. Another strategy to avoid the absorption of cholesterol, usually used as a complementary or an early preventive approach, which is better accepted by the general consumer, uses nutraceuticals, defined as food ingredients with pharmaceutical effects. There are several food ingredients that have the capacity to reducing cholesterol absorption (hypocholesterolemic effect), with health claims accepted by the European Food Safety Authority (EFSA) and the Food and Drug Administration (FDA), namely phytosterols (e.g., sterols and stanols), polyunsaturated fatty acids (e.g., Omega-3) and fibers (such as beta-glucans from oat, arabinoxylans from psyllium, galactomannans from guar gum, etc.), among others [15,16,17]. In common, all of them affect cholesterol bioaccessibility. 

Lipids such as sterols and stanols, along with the unsaturated and polyunsaturated fatty acids, are also known to be hypocholesterolemic [18]. In order to evaluate quantitatively how these classes of lipids affect cholesterol solubility in one of the most representative bile salts present in human intestinal lumen, glycodeoxycholic acid, an NMR assay methodology was developed, enabling the determination of cholesterol emulsified in these bile salt micelles directly [19]. This was an improvement regarding the previous indirect measurements, where performed chromatography or filtrations could perturb the equilibrium. These lipids had shown to act as co-solubilizers, competing for a place in the dietary bile salt mixed micelles, leading to a decrease of available cholesterol to be absorbed from intestinal lumen [20]. On the other hand, soluble fibers affect cholesterol absorption by different proposed mechanisms: (i) the increased viscosity in the intestinal lumen limits diffusion of hydrophobic molecules and their accessibility to the intestinal epithelium; (ii) interaction with cholesterol, with increased excretion in the feces; (iii) sequestration of bile salts, leading to a decrease in the emulsification of cholesterol and demoting the re-absorption of bile salts, thus promoting the conversion of cholesterol into bile salts in liver; and (iv) production of short chain fatty acids by intestinal bacteria that can regulate the production of endogenous cholesterol [21,22,23,24,25,26,27,28,29,30,31]. 

Considering the potential of unsaturated and polyunsaturated lipids and soluble fibers as hypocholesterolemic agents, the study of food matrices containing these compounds, such as coffee, could be important for the development of new and effective cholesterol-reducing ingredients. The coffee beverage is a rich source of fibers, accounting with 20% of the total soluble material [32], distributed between the presence of galactomannans (26%), arabinogalactans (20%), and the melanoidins (51%) which are complex structures formed during roasting by reaction of the previous polysaccharides with each other and with the protein and chlorogenic acids content existent in the coffee bean [33]. Several health-related properties have been attributed to the presence of fibers in coffee, including the immunostimulatory activity of galactomannans (GM) [34] and arabinogalactans (AG) [35,36] and several other health-related properties related to the presence of melanoidins [37,38,39]; however, to our knowledge, coffee fibers have not been explored regarding their hypocholesterolemic potential. Instead, most of the literature emphasizes the adverse effects upon coffee intake, such as the increase of cholesterol blood levels related to the diterpene content, critically dependent on the extraction method [40,41], or an increase of blood pressure due to coffee caffeine, a belief that is not a consensus in the scientific community [42]. The impact of coffee intake is even more important if consumed after the major daily meals, the optimal timing to affect cholesterol emulsification and absorption at the intestinal epithelium. In this work, selected espresso coffees and coffee extracts were used to address their effects on cholesterol solubility in an in vitro model composed by glycodeoxycholic bile salt, interpret their emulsification mechanisms and evaluate their cholesterol reducing potential, opening a new perspective on coffee consumption and its health impact. 

## 2. Materials and Methods 

### 2.1. Sample and Extracts Preparation

Coffee samples included single-dose coffee capsules, which were bought at national stores in Portugal. This study analyzed three different commercial blends, namely, a decaffeinated sample (S1) and two blend samples (S2 and S3). All espresso coffee samples (40 ± 2 mL) were prepared on a coffee pod machine, operating at a pressure of 19 bar, and extracted with distilled and deionized water. Details of several extracts and extraction conditions are in Table 1, based on their diversified content of arabinogalactans and galactomannans. 

Extracts E1 and E2 were selected from a dataset of cold-brew preparations [43]. They were obtained from solid–liquid extractions of freshly roasted grounded coffee (RGC) with distilled water, using different extraction conditions: extract E1 (10 min, 20 °C, 6 g) and extract E2 (360 min, 20 °C, 6 g), operating in Erlenmeyer flasks with extraction volume of 30 mL. The coffee ground used in the extraction of extract E1 was coarser than the one used in the extraction of E2. The flasks were cooled to room temperature, and their content was vacuum-filtrated through a glass microfiber filter (1.2 µm), and the filtrated content was frozen and freeze-dried prior to the analyses. The remaining soluble extracts were obtained by application of superheated water, under different operating conditions. Extract E3 (2 min heating with a heating rate of 50 °C/min plus 5.5 min at 120 °C under isothermal conditions, 2 g:60 mL) was prepared by using freshly grounded coffee, using microwave (MW)-assisted extraction conditions performed with a MicroSYNTH Labstation (Milestone srl., Bergamo, Italy), using 2 individual reactors, controlling the reactor temperature by the variation of the power of microwave irradiation [44]. The extract was then cooled to room temperature, and its content was vacuum-filtrated through glass microfiber filter (1.2 µm), and the filtrated content was frozen and freeze-dried prior to the analyses. 

Extract E4 (2 min heating with a heating rate of 90 °C/min plus 2 min at 200 °C under isothermal conditions, 2 g:60 mL) was obtained from spent coffee grounds (SCG), after beverage preparation, drying at 105 °C for 8 h, and application of microwave irradiation, as described elsewhere [45]. The water-soluble extract was then separated from the correspondent residue by centrifugation at 15,000 rpm, 4 °C for 20 min, and freeze-dried. To obtain richer polysaccharides fractions, the extract was further submitted to a purification step by ethanol precipitation. The extract was re-dissolved in the minimum amount of water, stirring during 10 min, at room temperature, and then absolute ethanol was added to reach aqueous solutions, respectively, containing 50% ethanol (*v/v*) and 75% ethanol (*v/v*) in a sequential order. The two precipitated fractions recovered were centrifuged at 15,000 rpm for 10 min at 4 °C. The 75% ethanol insoluble fraction to be used in this study was named extract E4. The Nescafé extract (E5) was prepared from an instant coffee powder commercial batch (Nescafé^®^, Dolce Gusto, Portugal), using the procedure as described before [35], briefly 15 g of sample was dissolved in 500 mL of water, at 80 °C, and stirred during 10 min, at the same temperature. After being filtered, the cold-water-soluble material with molecular weight higher than 100 kDa was purified at room temperature on an ultrafiltration module Labscale TFF System (Millipore, Jaffrey, New Hampshire, MA, America), using a Pellicon XL ultrafiltration Ultracel membrane (Merck Millipore, Jaffrey, New Hampshire, MA, America) with a cut-off of 100 kDa, working between 15 and 30 psi transmembrane pressures. The retentate material was frozen, freeze-dried and then weighted to prepare the solutions for solubility assays. The sugar composition of coffee extracts were determined as in previous works [32,35,45], and Table 2 show a summary of the mass fraction of galactomannans and arabinogalactans composition for the coffee extracts used in this work. The spent coffee grounds were obtained in a local cafeteria, after espresso-beverage preparation. The spent coffee grounds were dried and submitted to a Soxhlet solid–liquid extraction for 4 h, using n-hexane as solvent, to obtain a lipid fraction. The lipid composition was similar to those reported in the available literature [46]. 

Infusions of tea were obtained from two commercial brands available in a Portuguese commercial surface. Black Tea (BT) from Temeley^®^ and Aloysia Citrodora (AC) from Auchan^®^. The extraction was made by using 40 mL of preheated water from a coffee pod machine (using the same temperature of extraction than coffee) and using the commercial tea packets unidose submerged in water (infusion) for two minutes (as suggested by the tea suppliers). Colestipol hydrochloride was kindly supplied by Pharma 73 and was from Merck, and it was weighted in order to give a final concentration of 20 mg/mL. A total of 3 independent extractions were performed for each blend (40 ± 2 mL), and they were analyzed separately. 

### 2.2. Sugar Analysis

The coffee-brew extracts were weighed before and after lyophilization. The mass of sample used for sugar analysis was of 1 to 2 mg that were incubated with H_2_SO_4_ (72% *w*/*w*), at 30 °C during 3 h, with random stirring. The samples were then subjected to a 2 M solution of H_2_SO_4_, at 120 °C for 1 h, promoting the sugar hydrolysis [47].

The resultant free sugars were reduced with NaBH_4_ (15% in NH_3_ 3 M, 1 h, 30 °C) and further acetylated with acetic anhydride (3 mL), in the presence of 1-methylimidazole (450 μL), at 30 °C during 30 min. A extraction of alditol acetate derivatives from solution were done by washes with water anddichloromethane, and GC-FID analysis was performed by using 2-deoxyglucose as internal standard in a Perkin Elmer—Clarus 400 chromatograph (PerkinElmer, Massachusetts, USA), equipped with a FID detector and a DB-225 column (30 m × 0.25 mm and 0.15 μmof film thickness, J&W Scientific, Folsom, CA, USA). The temperature of the injector was 220 °C, while the detector operated at 230 °C. The following oven temperature program was used: initial temperature was set at 200 °C and then rose to 220 °C at 40 °C min^−1^, standing for 7 min, and reached 230 °C by a 20 °C min^−1^ rate, maintaining this temperature for 1 min. The carrier gas (H_2_) had a flow rate set at 1.7 mL min^−1^.

### 2.3. In Vitro Cholesterol Solubility Assay

Aqueous suspensions of ^13^C-4 cholesterol with and without different coffee samples or extracts were prepared by evaporating the required volume of a solution in chloroform/methanol (87:13, *v/v*), blowing dry nitrogen over the heated solution (blowing hot air over the external surface of the tube) and removing organic solvent traces in a vacuum desiccator for 30 min at 23 °C. Coffee lipid extracts in different concentrations were added, together with labeled cholesterol in the organic phase, and subjected to the latter described procedure. 

The glass tube with the dry residue was then transferred to a thermostatic bath at 37 °C and hydrated with a preheated solution of bile salts in the aqueous buffer, with a final concentration of 10 mM Tris–HCl (pH 7.4), 0.15 M sodium chloride (NaCl), 1 mM ethylenediaminetetraacetic acid (EDTA) and 0.02% sodium azide (NaN_3_) in 20% deuterated water (D_2_O). ^13^C-4 cholesterol and deuterium oxide (99.8%) for NMR experiments were obtained from CortecNet (Paris, France), the BS sodium sodium glycodeoxycholate (GDCA), and the 3-(Trimethylsilyl)propionic-2,2,3,3-d4 acid sodium salt (TSP) were obtained from Sigma (Steinheim, Germany). The organic solvents used for sample preparation (chloroform, methanol and hexane) were of spectroscopy grade, and the aqueous buffer components (Tris-HCl, NaCl, EDTA and NaN_3_) were of high purity and purchased from Sigma; water was first distilled and further purified by activated charcoal and deionization.

In order to achieve the buffer concentration, 120 μL of a buffer solution (5 times more concentrated) was added to 480 μL of coffee sample/extract, comprising the required volume (600 μL) for NMR measurements. The solubility assays of cholesterol for coffee extracts enriched in GM and AG designated by extract E1, E2, E3, E4 and E5 were performed, using mass volume ratio of 1 g/40 mL, in order to maintain the pattern found in espresso coffee [32]. The total concentration of bile salt was set at 50 mM in all solubilization experiments. The samples were left in the bath, with continuous stirring (100 rpm), during 24 to 48 h (experiments on the maximum solubilization), and then characterized by ^13^C NMR, to obtain the amount of solubilized cholesterol, using TSP as an internal standard [19].

Proton-decoupled ^13^C NMR spectra acquisitions were performed on a Bruker 500 MHz NMR spectrometer equipped with a high field ‘‘switchable’’ broadband 5 mm probe with z-gradient, operating at a frequency for ^1^H spectra (500.13 MHz) and for ^13^C spectra (125.8 MHz). Then, ^13^C NMR spectra were acquired at 37 °C, with a 30° pulse angle sequence, a spectral width of 25,252 Hz with an acquisition time of 1.3 s, a relaxation delay of 5 s and 2048 acquisition scans. Proton decoupling was achieved by using a WALTZ-16 decoupling sequence. After that, ^13^C-{^1^H} Nuclear Overhauser Enhancement (NOE) was obtained by comparing ^13^C spectra with full proton decoupling and ^13^C spectra with proton decoupling only during spectral acquisition, in a similar way as described elsewhere [19]. Spectra were processed with MestreNova 6.1.1 (Mestrelab Research, Santiago de Compostela, Spain). Then, ^1^H NMR spectra of were obtained at 37 °C with a 90° pulse angle sequence, a spectral width of 7500 Hz, acquisition time of 1.0 s, a relaxation delay of 5 s and 128 acquisition scans.

### 2.4. Determination of the Particles Size in Solution

The solutions after NMR measurement were filtered by using a syringe apparatus with a Millipore filter with a cutoff of 10 μm. The filtered solutions were analyzed by using dynamic light scattering to characterize the size of aggregates, using a Malvern Zetasizer Nano ZSP apparatus, operating with a He neon laser at 632.8 nm as light source and a scattering angle of 173°, using quartz cuvettes of 5 mm optical path. The autocorrelation curves were by fitted with up to three mono-exponential curves with the correspondent pre-exponentials and reciprocal characteristic decay time of each aggregate population present in solution. Determination of radius was obtained considering the Stokes–Einstein relationship for diffusion and using the relation between diffusion and the scattering vector obtained from the DLS measurements, as described elsewhere [48]. The parameters used for the calculations were the refraction index of 1.330, and the viscosity of 0.684 cp. The experiments were done at 37 °C, five measurements were run for each sample, and up to three independent samples were analyzed.

### 2.5. Determination of Crystals Thermotropic Behavior

The solids obtained after filtration were analyzed regarding its thermotropic behavior, using Polarized Light Thermal Microscopy (PLTM), linked to a hot-stage Linkam system, model DSC600, with a Leica DMRB microscope and a Sony CCD-IRIS/RGB video camera. Real-Time Video Measurement System software by Linkam was used for image analysis. The images were obtained by the combined polarized light and wave compensators, using a 200 magnification.

### 2.6. Statistics Analysis

Data were expressed as the mean and standard deviation of at least three independent experiments. Significant differences between samples in the parameter analyzed were determined by using the analysis of variance (ANOVA) with means separated with a multiple range test, Tukey’s range test, α = 0.05, using the Excel (Microsoft, Seattle, WA, America).

## 3. Results and Discussion

### 3.1. Sugars Analysis

The objective of this study was to identify if coffee has an effect on cholesterol bioaccessibility, as a measure of its hypocholesterolemic potential. In order to do so, a selection of coffee samples was performed, namely using different espresso coffee samples (S1–S3). Since carbohydrates are one of the most abundant components in coffee, potentially affecting cholesterol bioaccessibility, sugar composition was determined for coffee samples, as shown in Table 2. Additionally, a more diverse portfolio of coffee extracts was prepared, using alternative approaches, in order to reflect a major diversity/or maximize polysaccharide content: (1) using solid–liquid extracts, in order to allow different ratios of arabinogalactans and galactomannans, AG/GM content (see Table 2), with application of design of experiences in a broad range of conditions, from which extracts E1 and E2 were selected [43]; (2) application of pressurized conditions, attempting to increase the content of polysaccharides in solution, extracts E3–E5. In accordance with [49], a temperature of 150 °C interfaces the ratio of AG and GM, with extract E3 and E4, representing respectively a higher ratio of GM and AG. E4 was further submitted to an ethanol precipitation process that allowed an enrichment in polysaccharides, namely AG [50]. E5 is representative of an industrial pressurized extraction (instant coffee) sample, as the instant-coffee industrial preparation is a severe process which leads to the degradation of higher amounts of polysaccharides when compared to espresso-machine preparation, as this beverage contains a higher amount of low-molecular-weight material [35]. This sample was further enriched in arabinogalactans by application of ultrafiltration process. The sugar analysis from coffee espresso samples tested showed a composition rich in galactose, mannose and arabinose, with the decaffeinated sample S1 richer in galactose and S2 richer in mannose. The three coffee samples showed a similar total sugar composition; however, three different regimes of AG/GM ratios were obtained, namely a similar, lower and higher content. Because coffee samples are more complex in their composition, coffee extracts were addressed with different total sugar composition and with diverse AG/GM ratios, as seen in Table 2. 

### 3.2. Cholesterol Bioaccessibility and Bile Salt Sequestration

#### 3.2.1. Effect of Coffee Extracts

In order to better understand how coffee affects the emulsification processes taking place at the intestinal lumen, labeled ^13^C-4 cholesterol solubility was measured by ^13^C NMR [19,20], using an in vitro intestinal model containing bile salt micelles composed of glycodeoxycholic acid (GDCA), in the presence and absence of coffee blends of commercial origin (coffee capsules). The ^13^C NMR approach was chosen over ^1^H NMR because no distinctive proton resonances of cholesterol enable their quantitative quantification due to the molecular structure resemblance with GDCA. Moreover, due to the low isotopic abundance in ^13^C, a fully enriched ^13^C isotope of cholesterol, ^13^C-4 cholesterol, was used in these experiments, having a well-separated peak at 44 ppm chemical shift. The bile salt GDCA was chosen in these experiments due to its prevalence in intestinal lumen [51], in a concentration (50 mM) near the upper limit of bile salt physiological range in intestinal lumen (5–40 mM) [52]. Its higher solubility cholesterol index enables a higher sensitivity in the performed measurements [20]. Figure 1 shows a typical ^13^C NMR spectrum obtained with ^1^H decoupling and Nuclear Overhauser Effect (NOE), highlighting the 44 ppm region.

The use of NOE leads to an enhancement in the carbon signals, allowing us to abruptly decrease the number of scans needed for each experiment. However, as a drawback, the areas become no longer proportional to the concentration of the resonant carbon. To regain the quantitative power, a correction factor dependent on the carbon nature (C, CH, CH_2_, CH_3_) was determined by comparing the areas with and without NOE enhancement, as described in the methodology section and used in previous work [19]. With this methodology, cholesterol that is either precipitated as crystals or adsorbed to the fibers is not observed due to size, which influences the motion of these aggregates, causing the broadening of line width due to T2 (spin-spin) relaxation. The NMR methodology used enables us to follow cholesterol labeled in ^13^C that is emulsified in the micelles [19,20] and therefore the cholesterol that is available to be absorbed or bioavailable. Within the same measurement, bile salt content was also determined both by quantitative ^13^C NMR and crosschecked by ^1^H NMR, using TSP as internal standard, to evaluate the agreement between the quantification by ^13^C NMR, using NOE correction factors and ^1^H. The bile salt concentration variation gives information regarding the degree of sequestration of bile salts from solution.

The results show a cholesterol solubility decrease in the intestinal model used in the presence of three different commercial blends of espresso coffee from a Portuguese Commercial Roaster when compared with its absence. The analysis of cholesterol solubility was extended to a larger number of coffee extracts, using mild and harsh extraction conditions, namely hot water (E1 and E2) and subcritical water, obtained at pressurized conditions (E3–E5). These conditions were selected in order to evaluate the influence of the polysaccharides AG and GM ratio on the solubility of cholesterol. Moreover, to evaluate if this effect in bioaccessibility was specific of coffee or common to other food drinks, negative controls were used, consisting of two plant infusions, black tea (BT) and *Aloysia citrodora* (AC). A positive control, colestipol (identified as resin in Figure 2a), using a concentration of 20 mg/mL, was also tested to corroborate the approach followed. The overall results obtained for cholesterol solubility are shown in Figure 2a. 

The solubility of cholesterol is reduced in a range from 17% to 42% for all the coffee samples and extracts used. The infusions did not show a significant effect. The decrease of cholesterol solubility was more pronounced in espresso coffee samples than in coffee extracts. 

From the quantification of bile salt in solution, it was possible to observe that the concentration was 50.0 mM in the absence of coffee extracts, corresponding to the initial quantity of emulsifier used in the assays. However, in the presence of coffee samples and extracts, the concentration of bile salts decreases proportionally to the cholesterol solubility index, as shown in Figure 2b. 

These results show that the samples promote the decrease of bile salts’ concentration, leading to a decrease in the number of bile salts micelles available in solution for solubilizing cholesterol. The correlation obtained, although reflecting the variability and complexity of the coffee samples and extracts components, is meaningful for further evaluation of the coffee molecules that could be responsible for this effect. Based on this observation, and considering the hypothesis that bile salts are sequestered by coffee components from solution into aggregates that are undetected NMR due to their size [19,20], the study of the size of aggregates in solutions of the most representative coffee samples was performed. Figure 3 shows the results gathered from selected commercial coffee blends in the presence of bile salts micelles and cholesterol.

Aggregates with three different size magnitudes were found in the coffee samples’ solution: (i) 5 nm diameter, identified as bile salt micelles on the basis of previous analysis [53]; (ii) 140 nm; and (iii) a larger aggregate type with a size which varies for the different coffee samples analyzed. Coffee extracts analyzed did not show the presence of these bigger aggregates. The prevalence of aggregates of these sizes and the observation of reduction of bile salts’ concentration in solution reinforced the possibility of having bile salts sequestered. It is possible that the soluble fibers present in coffee, similarly to β-glucans, pectin and guar gum galactomannans [21,54], have a cholesterol-reducing effect. In order to test this hypothesis, the coffee extract E4 (see composition Table 2, results section), enriched in galactomannans (GM) and arabinogalactans (AG), was assayed in different concentrations. 

#### 3.2.2. Effect of Polysaccharides AG and GM Extracted from Coffee

Figure 4 shows the observed dependence of bile salt concentration (Figure 4a) and cholesterol solubility index (Figure 4b) on the [AG+GM] total concentration and its comparison with the content of these polysaccharides in the two datasets: the coffee samples (S1–S4) and extracts (E1–E5). A linear decrease of the bile salts concentration with the increase of [AG+GM] is observed for the different concentrations of E4 extract and for all the other coffee extracts (Figure 4a). However, for the coffee samples (S1–S4), no dependence of the bile salt concentration with the polysaccharides content is found. As observed for bile salts’ concentration, a similar reduction of cholesterol solubility index is seen with the increase of [AG+GM] concentration of the E4 extract (Figure 4b). 

Nevertheless, contrary to the dependence observed between bile salt solubility and polysaccharide concentration, the cholesterol solubility index is independent of the polysaccharide concentration, even for the coffee extracts (E1–E5), being an indication and reflecting that other compounds present in coffee matrix could be influencing the cholesterol solubility.

The decrease of bile salt solubility with the increase of polysaccharides content suggests that one of the mechanisms responsible for the decrease of cholesterol solubility is the coffee fiber sequestration of bile salts. However, this observation is not directly applicable to coffee samples because (1) coffee has aggregates of higher sizes than the extracts used (Figure 3b), (2) coffee is composed by a complex mixture when compared with extracts and (3) the variation of the fiber content does not influence the concentration of bile salts. Considering the dependence obtained for cholesterol and bile salts when using E4 extract (combining the data from Figure 4a and Figure 4b), it was possible to estimate the contribution of fibers present in the different coffee matrix samples. For example, for an average bile salts concentration of 40 mM, which corresponds to [AG+GM] concentration of 18 mg/mL (Figure 4a), the expected cholesterol solubility index is 2.1 mM, according to Figure 4b, corresponding to a reduction of 31%. This value is higher than the 1.5 mM observed for the coffee samples, showing a reduction of 46% in the cholesterol solubility index. This observation highlights that other compounds, besides fibers, can affect the coffee hypocholesterolemic potential, pinpointing the possible synergies between fibers and other coffee compounds. No clear relationship between the size of aggregates and their effect on cholesterol solubility or sequester capacity toward bile salts was found. 

To further investigate the possible existence of other mechanisms of cholesterol reduction, the thermotropic behavior of the cholesterol crystals, obtained from the solutions containing the coffee extracts in the intestinal model, was analyzed by using polarized light thermal microscopy. This technique was used to detect changes in crystal melting transition temperatures in those samples, which could indicate the presence of a co-crystallization mechanism. Typical photographs obtained for the crystals as function of temperature (Appendix A) show a melting transition at 147 °C, characteristic of the cholesterol anhydrous crystal phase transition [55]. This was observed for all the samples, and no significant differences were seen in that transition, ruling out the co-precipitation mechanism from the effects seen in the solubility of cholesterol. 

#### 3.2.3. Effect of Lipid Fraction Extracted from Coffee

Coffee samples, besides polysaccharides, also contain other compounds that could affect cholesterol solubility, such as lipids. The lipids content in the brew is highly dependent on the preparation method [46], and in an espresso coffee extract, it is below 0.37 mg/mL [56]. In order to evaluate the hypothesis that this amount could affect cholesterol solubility in bile salt micelles, a lipid fraction was tested in a range from 0.25 up to 2 mg/mL. The sample of lipids used was extracted from the spent coffee grounds, where the majority of the lipid fractions remains after beverage preparation, having a very similar composition to the coffee brew [45,54]. Lipid extracts obtained from coffee spent grounds showed a composition rich in unsaturated and saturated fatty acids, such as linoleic (45%) and palmitic (37%), respectively. Diterpenes account only for 5% of coffee lipids [57]. The mechanism by which these lipids affect cholesterol solubility could be the co-solubilization, as already described for saturated palmitic and unsaturated oleic acids, using the same approach as in the present work [20]. The effect of coffee lipid oil extracts on bile salt and cholesterol solubility index is shown in Figure 5. Bile salts concentration in solution does not vary with the lipid content (Figure 5a). However, the cholesterol solubility index is highly influenced by the lipids, showing a significant difference for a lipid concentration ≥0.25 mg/mL (Figure 5b).

These observations are compatible with a co-solubilization of lipids in the micelles, which leads to a smaller capacity to solubilize cholesterol, promoting its precipitation from the aqueous solution. The observation of the threshold minimum in cholesterol solubility suggests that bile salt micelles saturate with the coffee lipids and a further increase does not lead to changes in the cholesterol solubilizing capacity of the mixed micelles. As the concentration of the lipids used in these experiments is within the range found in espresso coffee [58], the expected cholesterol solubility effect promoted by the lipid fraction is a reduction from 3.2 to 2.5 mM. Considering that the contribution of the fibers for reduction of cholesterol is from 3.2 to 2.3 mM, both effects promoted by the espresso coffee lipids and the fiber, allowing an overall cholesterol reduction of 50%, from 3.2 to 1.6 mM. This value is similar to the average obtained for the cholesterol solubility index obtained for the coffee samples tested.

## 4. Conclusions

Coffee-water extracts used in this work led to the reduction of cholesterol solubility index in a bile salt dietary intestinal model. This was due partially to the presence of polysaccharides, as shown by the experiments with purified coffee extracts rich in galactomannans and arabinogalactans assayed in different concentrations. The mechanism by which these soluble fibers affect cholesterol solubility is related to their capacity to sequester bile salts from solution. Lipid extracts from coffee also affected cholesterol solubility. However, their behavior is different from the soluble fibers in two ways: (1) The bile salt concentration did not vary with the increased concentration of lipid, discarding the possibility of a bile salts sequestration mechanism; and (2) cholesterol solubility attains a threshold within the ranges of lipid concentration used. This is compatible with a co-solubilization mechanism, where lipid saturation of micelles is attained and no further solubilization of cholesterol is possible.

Polysaccharide and lipid effects on cholesterol bioaccessibility, measured by its cholesterol solubilization index, showed to be additive and representative of the overall effect observed for a typical coffee espresso.

This work is a contribution to the characterization of mechanisms of action and the quantitative determination of the effect of selected coffee extracted ingredients on cholesterol solubility in dietary micelles, leading to the establishment of rules to better describe and predict their behavior as possible hypocholesterolemic agents in the future.

## Figures and Tables

**Figure 1 nutrients-12-00437-f001:**
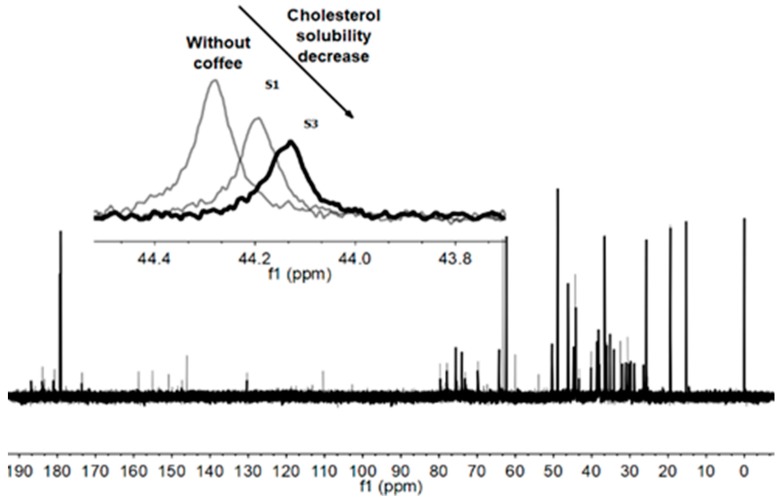
The ^13^C NMR spectra of 50 mM GDCA, with 3.5 mM ^13^C-4 cholesterol, in the presence of coffee extracts (S1 and S3 coffee pods). The spectra were acquired with ^1^H decoupling and NOE, in 10% D_2_O aqueous solution, at 37 °C. The peak of ^13^C-4 enriched cholesterol appears at 44 ppm. Note that the relative area of this peak decreases in the presence of coffee extracts, particularly for the S3 coffee extract (insert graph).

**Figure 2 nutrients-12-00437-f002:**
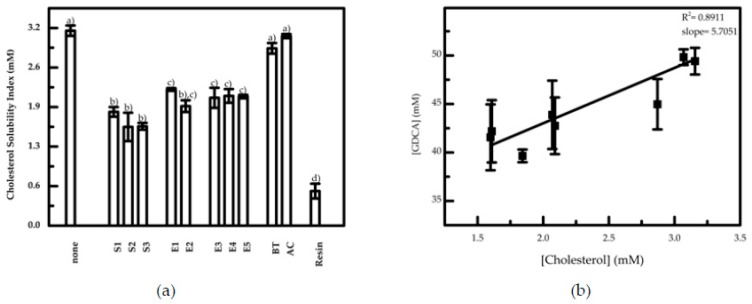
(**a**) Cholesterol solubility index obtained upon addition of commercial coffee, coffee extracts and tea infusions to an intestinal model made of GDCA bile salt micelles. Samples with the same character (a, b, c and d) represent values that are not significantly different (*p* < 0.05), when analyzing each parameter individually. (**b**) Cholesterol solubility dependence on bile salt concentration. The standard deviations presented corresponds to at least three independent experiments.

**Figure 3 nutrients-12-00437-f003:**
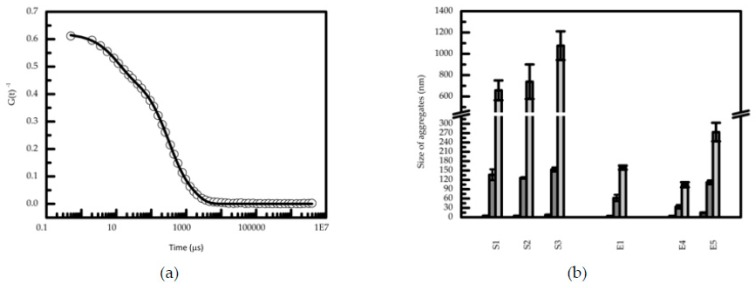
(**a**) Typical correlogram obtained for filtered (10 μm size) solutions, containing bile salt micelles with cholesterol and coffee sample (S2), at 37 °C. The line represents the best fit to three mono-exponential curves characteristic of the three aggregate species. (**b**) Size distribution of aggregates in selected coffee samples and extracts in the presence bile salts and cholesterol obtained by dynamic light scattering. The standard deviations presented corresponds to at least three independent experiments.

**Figure 4 nutrients-12-00437-f004:**
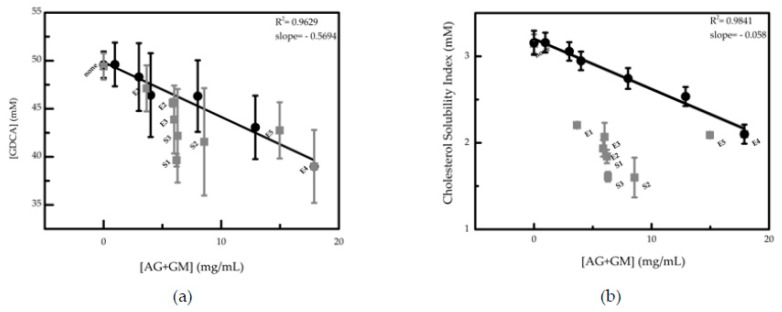
(**a**) Bile salt concentration and (**b**) cholesterol solubility index dependence on the total concentration of coffee galactomannans and arabinogalactans (filled black circles and linear fit). Coffee samples (S1–S3) and extracts (E1–E5) are represented as solid gray squares. The standard deviations presented correspond to at least three independent experiments.

**Figure 5 nutrients-12-00437-f005:**
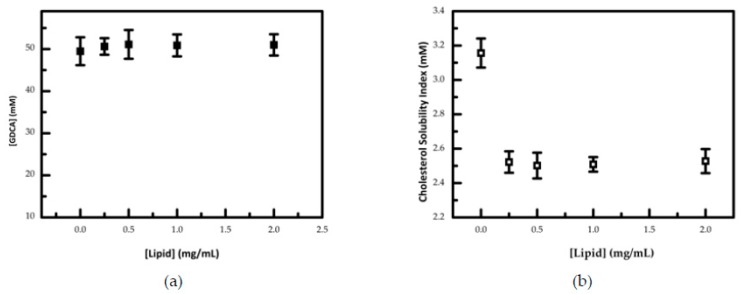
(**a**) Lipid effect on bile salt concentration. (**b**) Lipid effect on cholesterol solubility. The standard deviation presented is from at least three independent experiments.

**Table 1 nutrients-12-00437-t001:** Description of operating conditions regarding the coffee samples (S) and extracts (E).

Coffee Samples (S) and Extracts (E)		
Designation	Operating conditions	
S1 (Decaff)	Coffee pod machine extraction (19 bar, ratio 6 g: 40 mL)	
S2		
S3		
E1	Non-pressurized	RGC, Solid/liquid extraction (6 g: 30 mL, 20 °C, 10 min)
E2		RGC, Solid/liquid extraction (6 g: 30 mL, 20 °C, 360 min)
E3	Pressurized	RGC, MW (2 g: 60 mL, 2 min heating (50 °C/min) + 5.5 min (120 °C)
E4		SCG (Dry 105 °C/ 8h), MW (2 g: 60 mL, 2 min heating (90 °C/min) + 2 min (200 °C)
E5		IC, Solid/liquid extraction (15 g: 500 mL, 80 °C, 10 min)

**Table 2 nutrients-12-00437-t002:** Carbohydrate content and sugar composition of coffee samples (S) and coffee extracts (E). The polysaccharides composition of coffee samples estimated, using the known sugar composition for these samples [32,43]. The results shown are the average of at least three independent assays.

Sample Designation	Sugar Composition (% mol)					Total Sugars (g_sugar_/g_sample_)	Polysaccharides Content (g_polysaccharide_/g_sample_)		Ratio
Coffee Samples	Rha	Ara	Man	Gal	Glc		AG^a)^	GM^b)^	AG/GM
S1	6.2	19.5	22.1	45.5	6.6	0.25	0.16	0.06	2.6
S2	3.3	13.5	45.6	30.8	6.9	0.26	0.1	0.13	0.8
S3	5.3	16.9	35.8	29.6	12.4	0.24	0.11	0.1	1.1
Coffee Extracts									
E1	5.5	20.5	33.3	34.4	6.3	0.16	0.08	0.06	1.4
E2	3.7	20.1	41.4	29.7	5	0.22	0.1	0.1	1
E3	3.4	12.4	55.1	25.7	3.4	0.26	0.09	0.15	0.6
E4	0.0	12.6	18.7	66.8	1.2	0.67	0.52	0.14	3.8
E5	2.4	8.5	13.4	73.9	0.8	0.52	0.42	0.08	5.5

The estimation of the content of arabinogalactans (AG) considers all the amounts of arabinose and galactose at which it was subtracted the amount corresponding to a 5% of total mannose that is part of GM. The estimation also considers that all arabinose is part of AG. The polysaccharide content was determined by the following formulas: (a) $AG=\frac{\;\left(Ara+Gal-0.05*Man\right)}{100}\times \frac{g_{sugar}}{g_{sample}};$ (b) $GM=\frac{\;\left(Man+0.05*Man\right)}{100}\times \frac{g_{sugar}}{g_{sample}}$.

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
