# Peer review of "In Vitro Hypocholesterolemic Effect of Coffee Compounds"

_nutrients, 2020, doi:10.3390/nu12020437_

Round 1

Reviewer 1 Report

This manuscript is of interest. The authors used an interesting model to demonstrate in vitro hypocholesterolemic effects of coffee. The manuscript is very well written and conclusions are supported by the results. I have only a request, the cultivar of the espresso coffe used is to be reported.

Author Response

Reviewer Comments:

"This manuscript is of interest. The authors used an interesting model to demonstrate in vitro hypocholesterolemic effects of coffee. The manuscript is very well written and conclusions are supported by the results. I have only a request, the cultivar of the espresso coffe used is to be reported."

Authors answer:

We would like to acknowledge the reviewer positive comments. Regarding the suggestion made, unfortunately the authors were not able to get the information of coffees cultivar due to the commercial nature of the coffee. Nevertheless, the espresso coffee pods are representative of the usual coffee intake by regular consumers.

Reviewer 2 Report

Title: In vitro hypocholesterolemic effect of coffee compounds

Daily coffee consumption has many proved beneficial effects of human health. However, the relationship between coffee and cholesterol is controversial. Although most studies have noted that filtered coffee has a neutral effect on lipid levels, unfiltered coffee appears to increase LDL, total cholesterol, and triglycerides in some studies. Diterpenes found in high amounts in unfiltered coffee, cafestol and kahweol, have been found to raise cholesterol levels.

This research paper pretends to clarify the effect of specific compounds present in coffee beverages on cholesterol bioaccessibility. The aim of this study was to identify if coffee has effect on cholesterol bioaccessibility, as a measure of its hypocholesterolemic potential. The effect of commercial espresso coffee and coffee extracts on cholesterol solubility were studied in an in vitro model composed by glycodeoxycholic bile salt. In general, I found the paper to be well written and much of it well described. The clarification of the effect of coffee on cholesterol bioaccesibility is novel and of great interest to the global population. I have a few comments regarding specific details of the investigation and I kindly ask the authors to answer to them.

It has been reported that individuals consuming roughly 60 milligrams of cafestol (equivalent to 10 cups of unfiltered, French press coffee or 2 grams of coffee oil) may raise total cholesterol levels by an average of about 20%. This is largely due to an increase in low density lipoprotein (LDL) levels and triglyceride levels. High-density lipoproteins (HDL) do not appear to be affected. It is thought that filtered coffee does not have this effect because the diterpenes are caught in the filter and not included in the coffee consumed. It seems like the authors filtered all samples during their preparation. Did they consider studying the effect of non-filtered coffee? Minor comments: Please check English grammar, most of the passive forms are not correctly written. Units of total sugars are missing in Table 2. I suggest making Figure 1 bigger. Reference number 47 is missing volume and issue numbers.

Author Response

Reviewer 2 Comment 1:

"It has been reported that individuals consuming roughly 60 milligrams of cafestol (equivalent to 10 cups of unfiltered, French press coffee or 2 grams of coffee oil) may raise total cholesterol levels by an average of about 20%. This is largely due to an increase in low density lipoprotein (LDL) levels and triglyceride levels. High-density lipoproteins (HDL) do not appear to be affected. It is thought that filtered coffee does not have this effect because the diterpenes are caught in the filter and not included in the coffee consumed. It seems like the authors filtered all samples during their preparation. Did they consider studying the effect of non-filtered coffee?" 

Authors response:

We acknowledge the Reviewer to highlight the point regarding the diterpenes content in unfiltered coffee and their relation with high blood cholesterol levels. In fact, the manuscript introduction already referred the hipercholesterolemic effect of coffee diterpenes, modulated by the extraction methodologies.  The lipid fraction used in this work, described in the methodology, contains 0.12 mg/mL diterpene in the higher concentration used  (this corresponds to 5 mg per 40 mL coffee drink, which is roughly equivalent to the 6 mg per cup mentioned by the Reviewer). Nevertheless, diterpenes account only for 5 % of coffee lipid fraction, the reason why they should not have a relevant influence in the emulsification of cholesterol modulated by coffee lipids. This information was now include in section 3.2.2. This does not exclude the fact that other mechanisms can be responsible for the reported hipercholesterolemic effect  related to diterpenes content, such as the downregulation of cholesterol 7 alpha-hydroxylase  (already shown in rat hepatocytes), supressing the bile acid synthesis and affecting cholesterol homeoastasis in the liver.

Reviewer 2 Comment 2

"Minor comments: Please check English grammar, most of the passive forms are not correctly written. Units of total sugars are missing in Table 2. I suggest making Figure 1 bigger. Reference number 47 is missing volume and issue numbers."

Authors response:

The following changes (highlighted in red) were made: 

  i) The passive forms, when possible, were avoided. 

 ii) The units of total sugars were added in Table 2.

iii) Figure 1 was modified.

iv) The reference number 47 was updated with the volume and issue numbers.